ecology, behaviour

*Acyrthosiphon pisum*, colony, colour mutant, density-dependent effect, reproductive restraint

**Author for correspondence:**
Shin-ichi Akimoto
e-mail: akimoto@res.agr.hokudai.ac.jp

# Self and non-self recognition affects clonal reproduction and competition in the pea aphid

Yang Li[1,2] and Shin-ichi Akimoto[1]

[1]Department of Ecology and Systematics, Graduate School of Agriculture, Hokkaido University, Sapporo 060-8589, Japan
[2]College of Biology and Agriculture, Zunyi Normal University, Zunyi Guizhou 563006, People's Republic of China

YL, 0000-0001-7955-6009; S-iA, 0000-0002-7038-9678

The spatial interaction of clonal organisms is an unsolved but crucial topic in evolutionary biology. We evaluated the interactions between aphid clones using a colour mutant (yellow) and an original (green) clone. Colonies founded by two aphids of the same clone and mixed colonies, founded by a green aphid and a yellow aphid, were set up to observe population growth for 15 days. We confirmed positive competition effects, with mixed colonies increasing in size more rapidly than clonal colonies. In mixed colonies where reproduction started simultaneously, green aphids overwhelmed yellow aphids in number, and yellow aphids restrained reproduction. However, when yellow aphids started to reproduce earlier, they outnumbered the green aphids. To test whether aphids have the ability to control reproduction according to the densities of self and non-self clones, one yellow aphid or one antennae-excised yellow aphid was transferred into a highly dense green clone colony. Intact yellow aphids produced fewer nymphs in crowded green colonies, whereas the fecundity of antennae-excised aphids did not change. Thus, we conclude that aphid clones can discriminate between self and non-self clones, and can regulate their reproduction, depending on whether they are superior or inferior in number to their competitors.

## 1. Introduction

Clonal reproduction is the major mode of propagation for some plants, parasites, marine benthic animals and aphids. In clonal organisms, competition among clones shapes evolutionary outcomes [1–5]. Several empirical studies have demonstrated that competition among unrelated clones is so strong that it leads to a great reduction in clone diversity within a local population [6–10]. In addition to intense competition among unrelated clones, clonal organisms reportedly often have the ability to discriminate between kin and non-kin clones, and to control the strength of competition depending on their relatedness to the neighbouring clone [1,11–14]. An increasing number of studies have documented that, as with other immobile clonal organisms, plants are capable of self–non-self discrimination and can adjust root and shoot growth, depending on the identity of the neighbouring plants [15–18].

Aphids (Aphididae; Insecta) are peculiar among clonal organisms in that clone members are independent of each other and mobile [19]. In the wild, different aphid clones often coexist, forming mixed colonies on a host plant [20–24]. Clonal mixing can lead to competition among aphid clones over nutrition and space on the same host plant [25–27]. Furthermore, clonal mixing can hinder the evolution of altruistic behaviour such as soldier production [23,28]. Considering the role of aphids as important crop pests, it is an evolutionarily and practically crucial issue to understand whether aphid clones can

discriminate between closely related and unrelated clones, and how they regulate their reproduction in mixed colonies. Nevertheless, very few studies have addressed this issue [20,26,27,29]. Regarding aphid species that produce a solder caste or induce communal galls, it has been reported that they have no ability to discriminate between kin and non-kin members [20,30,31]. However, it is not known whether this feature applies to non-social aphid species. Some authors have compared reproductive rates, distribution patterns on plants, and reactions to parasitoids between clonal colonies and mixed colonies of aphids [26,27,29], confirming the existence of significant interactions between clones in some cases. Mixed colonies are reported to reproduce faster [26,27] and exhibit a different response to parasitoids than clonal colonies [29]. However, these studies used DNA markers to distinguish clones, which has the disadvantage of precluding the observation of detailed interactions between live aphid clones. Furthermore, the rearing of aphid colonies on growing plants makes it difficult to precisely count aphid numbers throughout the rearing period.

To overcome these disadvantages, we used a colour mutant and its original clone of the pea aphid, *Acyrthosiphon pisum*, and employed the agar-leaf rearing method [32] to separately evaluate the daily reproduction of the different clones. *Acyrthosiphon pisum* is a serious pest aphid that feeds on leguminous plants and is distributed worldwide [33]. The two clones were so distinct in colour so as to discriminate clones in all stages. Because we reared the aphids on cut leaves attached to agar medium, we were able to evaluate daily reproduction of the respective clones for 15 days, as well as the distribution patterns of the clones on the leaves.

In the present study, we tested the hypothesis that aphids can discriminate self and non-self clones, and can alter their reproduction accordingly. For this purpose, we evaluated (1) the population growth of each clone in colonies founded by a single aphid, two clonal aphids, and two non-clonal aphids (i.e. a mixed colony), (2) the population growth in mixed colonies where two aphid clones start to reproduce simultaneously or with different timing, (3) the reproductive rate of an antennae-equipped or antennae-excised aphid when it was transferred to highly dense self or non-self colonies and (4) the distribution pattern of the members of each clone in a mixed colony to understand whether the clones avoid each other. Synthesizing the results from these experiments, we will, for the first time, indicate that aphids can discriminate self and non-self clones and regulate their reproduction depending on whether they are surrounded by their own clonal members or by members of a distinct clone.

## 2. Material and method

### (a) Study organisms

We used two *A. pisum* clones for our experiments, one of which (CR13028c) originated from laboratory crosses, and the other being a colour mutant (CR13028c-Y) derived from the former. CR13028c was a backcross strain, in which sexual females from a hybrid clone (paternal origin: a *Vicia cracca*-associated clone from Hachinohe, Japan; maternal origin: a *V. sativa* subsp. *nigra*-associated clone from Hachinohe) were backcrossed to males from the *V. cracca*-associated clone. The colour mutant appeared in the stock culture of the original clone in spring, 2017, and had been reared separately. Thereafter, all aphids that were produced

parthenogenetically exhibited yellow body colour (hereafter referred to as the yellow clone), which contrasted well with the green body colour of the original clone (hereafter referred to as the green clone) (electronic supplementary material, figure S1). This colour difference enabled us to count the two clones separately within mixed colonies and, thus, the number of newborns daily. In the experiments, apterous adults that were reared at a low density were used for both clones.

### (b) Competition between the clones

To evaluate population growth patterns in the yellow and green clones, we transferred fourth instar aphids onto broad bean leaves, allowed them to reproduce, and counted the total number of aphids, daily, for 15 days (from the first day of larviposition (day 1) to day 15). For rearing, we used the agar-leaf method [32], in which aphids were transferred onto cut leaves on agar medium containing nutrient solution. In this system, aphids grow and reproduce on cut leaves as successfully as they do on broad bean seedlings. Three treatments were prepared, the first of which was a single aphid treatment, which started with one aphid of each clone (1G or 1Y) being transferred onto a leaf. To evaluate density-dependent effects among clonal members, the two-aphid treatment was created by transferring two aphids of each pure clone (2G or 2Y) onto different leaves. In the mixed clone treatment, two aphids of different clones (1G + 1Y) were simultaneously transferred onto different leaves to test whether the coexistence of different clones leads to competitive interactions. For each treatment, 10 replicates were prepared.

In the mixed clone treatment, it is likely that a clone with high reproductive rate is dominant in the competition. However, the outcome of the competition may be affected by the timing of reproduction [34]. In the mixed clone treatment, we additionally prepared experiments where a yellow aphid reproduced 2 or 3 days earlier than a green aphid to test whether the timing of reproduction affects the outcome of competition. For these experiments, the final colony size at 16 days after the transfer of yellow adults (and 12 or 13 days after the transfer of green adults) was determined. We examined how the initial difference in the numbers of yellow and green nymphs affect the final sizes of these clones using all experiments of the simultaneous and time lag installation.

Round plastic containers (10 cm diameter and 5 cm height) containing agar medium (3 cm height) were used for all of the experiments. A new leaf was placed on the agar surface every 4 days. During observation, if any foundress died, the replicate was removed and a new replicate was prepared. All the containers were placed in a climatic chamber (MIR-254; Sanyo Corporation, Mito, Japan) that was set to 20°C, 50–60% relative humidity, under a 16 L : 8 D photoperiod at 5.8–7.3 W m$^{-2}$.

### (c) Reproduction in highly dense colonies

We tested whether an aphid's reproductive activity changes when it is placed in highly dense colonies of self or non-self clones. A yellow adult prior to larviposition was transferred to a colony of 130 to 150 aphids consisting of either self or green clone, and the aphid's fecundity was evaluated for 3 days after larviposition. The highly dense colonies were prepared by transferring five aphids of the same clone onto one leaf as foundresses and allowing them to reproduce freely for 5 days. Another leaf was added 3 days later. When a yellow adult was transferred to a highly dense colony of its own clone, it was impossible to distinguish nymphs produced by that aphid and by other aphids. Therefore, all adults were removed from the colony before transfer and a young adult was subsequently transferred into the colony so that we were able to guarantee that newborns were produced by the transferred aphid.

To clarify how an aphid detects self and non-self clones, for some yellow adults, we excised both antennae at the third antennal segment using fine forceps. Antennae-excised yellow aphids were then individually transferred to a highly dense colony of the green clone or of self clone to observe reproduction for 3 days.

## (d) Spatial distribution of the two clones

We expected that, if the different clones avoided each other, non-clonal members would be situated more distantly from each other than from clonal members. To examine this, we used images of the mixed colonies from day 12, when the difference in population size between the clones became obvious (see results). The positions of all aphids were located within an X–Y coordinate configuration using ImageJ [35], with the clones being discriminated. We measured the distances between all possible pairs of aphids and between all members of the same clone (yellow or green) and compared the mean distance among green–green pairs (G–G), yellow–yellow pairs (Y–Y) and green–yellow pairs (G–Y). The number of distances less than 5 mm (short distance) and distances $\geq 5$ mm (long distance) were then counted for clonal and non-clonal members. For the measurement and categorization of the distances, the R functions 'dist' and 'hist' were used.

## (e) Statistics

Differences in population growth pattern were tested with repeated-measures MANOVA [36] because its objective variable, colony size, involved multiple measures of the same colony over 15 days. The statistical significance of within-subject interactions between days and clones (or colonies under different conditions) was checked to determine whether or not the population growth patterns were different. When the assumption of sphericity was satisfied for all possible pairs of days (i.e. the equivalence of the variance of difference in colony size between all possible pairs of days), the interaction would be tested with $F$ statistics. However, if this assumption was violated, then the degrees of freedom were modified such that a valid $F$ ratio could be obtained. Our analyses showed that the assumption of sphericity was violated in all cases, and so we used the Greenhouse–Geisser correction [35] to adjust the degrees of freedom. When population growth was compared between colonies founded by a single foundress and colonies founded by two clonal foundresses, the population size of the latter on each day was divided by two to obtain the per capita growth pattern. The relationship between the timing of reproduction by the two clones and the final proportions of the clones was examined by the generalized linear model (GLM) with a binomial error structure. In the transfer experiments, differences among treatments in the number of nymphs produced over 3 days were tested using the Tukey–Kramer method. These statistical analyses were performed using JMP v. 13 (SAS Institute, Cary, NC, USA).

Regarding the distribution patterns of aphids on leaves in mixed colonies, the mean distances between aphids were compared among G–G pairs, Y–Y pairs and G–Y pairs using ANOVA. Differences in the number of short distances relative to that of long distances were also tested among the three groups using the generalized linear mixed model (GLMM). In the model, aphid density in each container was included as a covariate and variation among containers was treated as a random effect. The analysis was conducted using the glmmML function in the package 'glmmML' in R v. 3.2.1 [37], with a binomial error structure.

## 3. Results

In all of the treatments, the second generation that was produced on day 1 started reproducing on day 10, resulting in a steeper colony growth curve after day 10 (figure 1). Colonies founded by a single green aphid (1G) increased more rapidly in number than those founded by a single yellow aphid (1Y) (figure 1a; within-subject interaction between time and clone, d.f. = 1.68,30.23, $F = 6.49$, $p = 0.0066$). Similarly, colonies founded by two green aphids (2G) increased more quickly than those founded by two yellow aphids (2Y) (figure 1b; within-subject interaction between time and clone, d.f. = 2.84,51.04, $F = 7.56$, $p = 0.0004$), but no significant difference was detected in final colony size between the two clones (ANOVA, d.f. = 1,18, $F = 0.21$, $p = 0.65$). The final colony sizes for 2G and 2Y colonies were on average 1.17 and 1.35 times that of 1G and 1Y colonies, respectively.

Mixed colonies (1G + 1Y) increased more rapidly and attained a larger final size than colonies founded by two clonal aphids (within-subject interaction between time and clone, 1G+1Y versus 2G, d.f. = 2.71,48.86, $F = 9.35$, $p < 0.0001$; 1G+1Y versus 2Y, d.f. = 2.58,46.47, $F = 9.22$, $p < 0.0001$; ANOVA on day 15, 1G+1Y versus 2G, d.f. = 1,18, $F = 80.7$, $p < 0.0001$; 1G+1Y versus 2Y, d.f. = 1,18, $F = 49.2$, $p < 0.0001$). The final sizes of mixed colonies were 1.14 and 1.13 times that of 2G and 2Y colonies, respectively (for mixed colonies versus 2G, d.f. = 1,18, $F = 80.7$, $p < 0.0001$; for mixed colonies versus 2Y, d.f. = 1,18, $F = 49.2$, $p < 0.0001$). In mixed colonies, green aphids overwhelmed yellow aphids in number (figure 1c,d; within-subject interaction, d.f. = 2.13,38.27, $F = 96.89$, $p < 0.0001$), and yellow aphids largely restrained their reproduction after day 9 (figure 2). On day 11, the mean numbers of green and yellow aphids in mixed colonies were 88.4 ($\pm 5.52$ s.e.) and 61.8 ($\pm 2.76$), respectively. The difference negatively affected the production of newborns by yellow aphids the next day. On day 12, yellow aphids produced five times fewer nymphs ($8.0 \pm 1.24$) than did green aphids ($40.9 \pm 4.72$) (ANOVA, d.f. = 1,18, $F = 45.46$, $p < 0.0001$).

We compared *per capita* reproductive rates between the two-clonal aphid treatment and the mixed clone treatment (figure 1c,d). The green aphid reproduced more rapidly when it coexisted with a yellow aphid (1G in 1G + 1Y) than when it coexisted with a fellow clone (1G in 2G) (figure 1c; within-subject interaction between time and clone, d.f. = 2.06,37.01, $F = 49.61$, $p < 0.0001$). Colonies founded by 1G alone increased more rapidly than 1G in 2G colonies (figure 1c; within-subject interaction, d.f. = 1.79,32.28, $F = 87.65$, $p < 0.0001$) and 1G in 1G+1Y colonies (d.f. = 1.77,31.82, $F = 3.64$, $p = 0.0425$). Regarding yellow aphids, since they were outnumbered by green aphids in mixed colonies, their per capita reproduction in mixed colonies (1Y in 1G + 1Y) was much lower than that in the 2Y treatment (1Y in 2Y) (figure 1d; within-subject interaction, d.f. = 2.64,47.51, $F = 33.07$, $p < 0.0001$). Nevertheless, the yellow aphid produced a larger number of nymphs by day 10 when it coexisted with a green aphid than when it coexisted with a fellow clone (d.f. = 3.07,55.24, $F = 15.57$, $p < 0.0001$). Colonies founded by 1Y alone increased more rapidly than 1Y in 2Y colonies (figure 1d; within-subject interaction, d.f. = 1.96,35.35, $F = 41.93$, $p < 0.0001$) and 1Y in 1G+1Y colonies (d.f. = 1.97,35.38, $F = 91.40$, $p < 0.0001$).

When yellow adults were manipulated to start reproduction 2 or 3 days earlier than green adults, yellow aphids outnumbered green aphids (figure 3a: minus numbers in the horizontal axis mean the numbers of pre-existing green nymphs). If a yellow adult had produced a larger number of nymphs prior to the competitor's reproduction, yellow aphids accounted for higher proportions in the final mixed

**Figure 1.** Population growth of colonies (cumulative number of aphids) founded by a single aphid, two clonal aphids and two non-clonal aphids over 15 days (mean ± s.e.). For each curve, $n = 10$. (*a*) Colonies founded by a single aphid (upper: 1G, lower: 1Y); (*b*) colonies founded by two clonal or two non-clonal aphids (upper: 1G + 1Y, middle: 2G, lower: 2Y); (*c*) colonies founded by a single green aphid or a green aphid coexisting with a yellow or a green aphid (upper: 1G, middle: 1G coexisting with 1Y, lower: 1G coexisting with 1G); (*d*) colonies founded by a single yellow aphid or a yellow aphid coexisting with a yellow or a green aphid (upper: 1Y, middle on the final day: 1Y coexisting with 1Y, lower on the final day: 1Y coexisting with 1G). 1G = one green aphid; 2G = two green aphids; 1Y = one yellow aphid; 2Y = two yellow aphids.

colonies; logistic regression predicted that if on average 5.5 and 18.9 yellow nymphs pre-exist in the mixed colony, then the final proportion of yellow aphids account for 50.0% (48.8–51.0%; 95% CI) and 75.0% (73.5–76.4%), respectively. Yellow aphids produced on average 10.9 (±3.33 s.d.) and 15.2 (±4.15) nymphs for the first 2 and 3 days, respectively. In addition, there was a link between the proportion of yellow aphids in a mixed colony and the final colony size (figure 3*b*). The final size of mixed colonies was larger when the proportion of yellow aphids was intermediate (45–71%) than for smaller or larger values (AIC for quadratic regression 268.4, AIC for linear regression 275.1). This result shows that aphids continue to reproduce when they are not inferior in number to their competitors.

When a single yellow aphid was transferred to a highly dense colony of the green clone, it produced a significantly smaller number of nymphs than did the control (1Y) (figure 4). When a single yellow aphid was transferred to a highly dense colony of its own clone, its fecundity did not change compared to that of 1Y. Interestingly, when an antennae-excised yellow aphid was transferred to a high-density colony of the green

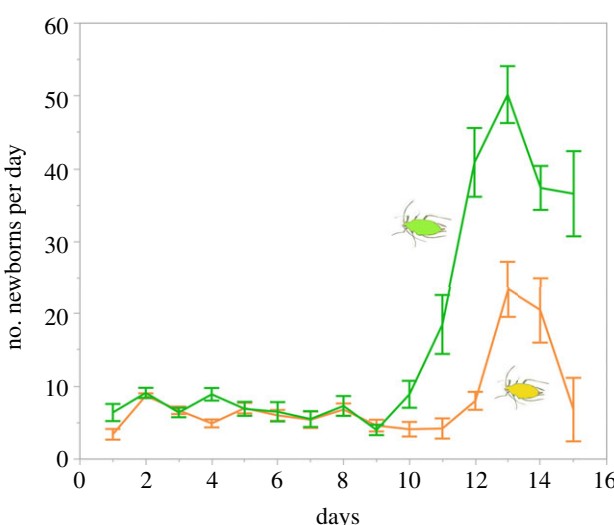

**Figure 2.** Daily production of newborns by the green (upper) and the yellow (lower) clone in mixed colonies (mean ± s.e.). For each curve, $n = 10$. (Online version in colour.)

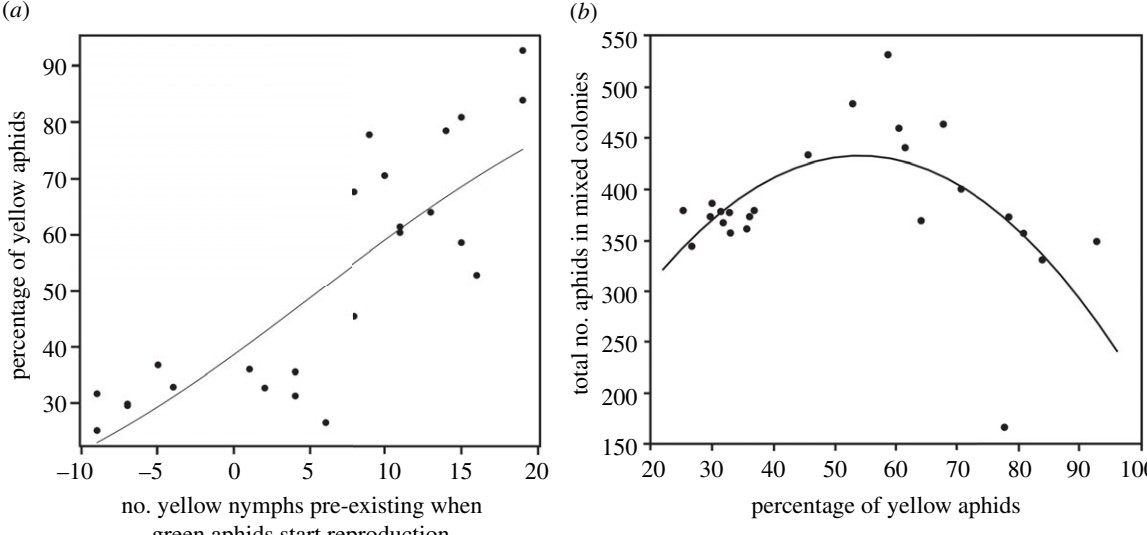

**Figure 3.** Relationship (*a*) between the timing of reproduction of the two clones and the final percentage of yellow aphids in mixed colonies and (*b*) between the final percentage of yellow aphids and the total number of aphids in mixed colonies. The horizontal axis in (*a*) indicates the number of yellow nymphs pre-existing when green aphids start reproduction. Negative numbers represent the numbers of green nymphs pre-existing. (Online version in colour.)

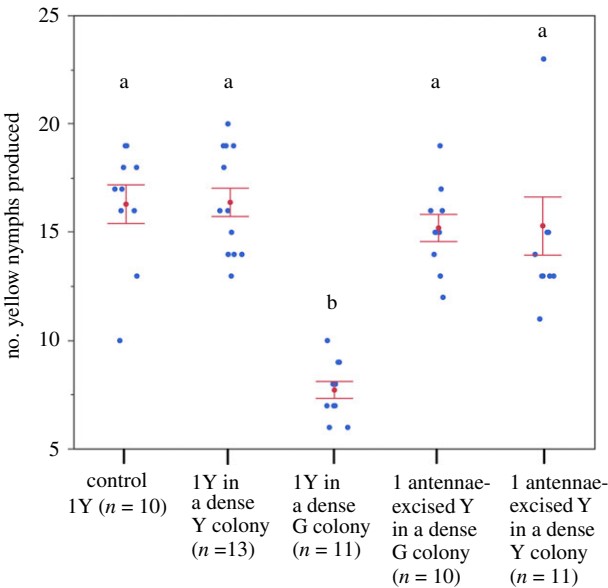

**Figure 4.** Comparison of the number of nymphs produced by a single yellow aphid for the first 3 days after larviposition (mean ± s.e.). Comparison was made among a control aphid and a single aphid transferred to a highly dense colony of the green clone (G) or the yellow clone (Y). The comparison included an antennae-excised yellow aphid transferred to a highly dense colony of the green or yellow clone. (Online version in colour.)

clone, its fecundity was not significantly different from that of 1Y, suggesting that the aphid was not able to perceive the presence or crowding of the different clones. The same was true for an antennae-excised yellow aphid that was transferred to a high-density colony of the fellow clone.

The mean distance and the percentage of distances less than 5 mm between aphids for each of G–G pairs, Y–Y pairs and G–Y pairs in each replicate are shown in the electronic supplementary material, table S1 and figure S2. G–Y pairs were located more distantly than G–G pairs or Y–Y pairs in every cage (Tukey–Kramer test: $p < 0.05$). The percentage of G–Y pairs situated within short distances was significantly smaller than that of G–G pairs (GLMM, $z = 7.54$, $p < 0.0001$), but not significantly different from that of Y–Y pairs

($z = 1.87$, $p = 0.0618$). Aphid density had no significant effect on the percentage ($z = -1.42$, $p = 0.154$).

## 4. Discussion

Using a colour mutant and a novel rearing method, we were able to examine the competition dynamics between and within aphid clones. Remarkably, we found that an aphid can regulate its reproductive rate depending on whether it coexists with a clonal or non-clonal member. When a colony starts from two clonal members, their reproduction is self-regulated from the initial founding stage, and this birth control becomes more prominent in the subsequent generation. By contrast, in the case of an aphid coexisting with a non-clonal aphid, no density-dependent control of reproduction exists at the initial stage, leading to moderate but significant competition effects. The positive effects of competition in mixed colonies, that is, higher reproductive rates, have also been reported in other aphids [26,27], as well as in *Chlamydomonas* and malaria parasites [38,39].

Competing clones exhibited different reproductive patterns after the onset of reproduction by the offspring generation. When a given clone outnumbered its competitor, it maintained high reproductive rates. However, when the clone was outnumbered by the competitor, it restrained its reproduction by a considerable degree (figures 2 and 4), suggesting that the clone can evade competition with little chance of winning. As a result, competition between the different clones resulted in a clear-cut winner and loser; the green clone far exceeded the yellow clone in the final colony size (2.1 : 1) compared to the ratio of the final colony size of 1G and 1Y colonies (1.1 : 1). These results indicate that aphid clones can regulate their reproductive rate in competitive situations.

The results of the experiments manipulating birth timing corroborated this observation. The outcome of clone–clone competition was reversed when yellow aphids started reproduction a little earlier than green aphids, and this finding indicates that the outcome of competition is affected by the relative timing at which the two clones start reproduction as well as by their reproductive rates.

**6**

Manipulation of birth timing also indicated that where the two clones did not recognize their numerical superiority, the regulation mechanism for reproduction did not work, leading to high densities in both clones (figure 3b). These results suggest that aphids have a flexible ability to regulate their reproduction according to the relative density of self and non-self clones.

Given the ability to regulate reproduction, natural selection would favour not only aphids with higher reproductive rates but also those that can access food resources earlier. For example, the timing of egg hatching could be subject to intense selection because foundresses hatching earlier in spring can access preferred resources and start reproduction earlier. Several studies have reported instances of intense competition among nymphal foundresses [40,41]. Similarly, aphid clones that tend to produce a high proportion of winged females can access unused host plants earlier, albeit with a low reproductive rate. The ability of aphids to access resources earlier could compensate for low reproductive rates, possibly leading to a fecundity–dispersal trade-off [42].

The transfer experiment of single aphids corroborated the hypothesis of reproductive restraint only for the yellow clones. We observed that reproductive activities changed significantly between yellow aphids that were transferred to highly dense green colonies and those that were transferred to highly dense fellow colonies. A high density of the green clone has a stronger negative effect on the reproduction of the yellow clone than that of fellow clonal members. The reason for the difference in reproductive activities may be due to the relatedness of colony members. Where an adult is placed in a fellow colony, all aphids are clonal, so that they may behave altruistically, sharing food resources. However, where an adult is surrounded by members of non-self clone, different clones probably behave selfishly such that sharing of food resources may be limited, resulting in lower viability or performance of its offspring. In this situation, reproductive restraint might be advantageous. Although the present study was not able to determine the fate of defeated clones, in the wild, defeated clones probably moved to different parts of the host plant or to different plants to resume reproduction [25,43].

There was evidence to suggest that the yellow clone was a dominant mutant (electronic supplementary material, table S2) and had different phenotypes from the green clone with regard to body colour and reproductive rates. The hypothesis that competing clones could detect the presence of each other through the antennation was tested in the present study by the antennae-excision experiment. The fact that the 3-day fecundity of an antennae-excised yellow aphid that was transferred to a highly dense green colony was not significantly different from that of a control yellow aphid suggests that some substances that can be detected by antennation are involved in the recognition of self and non-self clones or crowding. Comparisons of 3-day fecundity between intact and antennae-excised aphids in highly dense fellow colonies suggest that antennae excision had negligible effects on their reproduction. The body surface of aphids is covered with cuticular hydrocarbons [44], which are used by the attending ants and predators as cues [44,45]. Similarly, it is likely that aphid clones partially use cuticular hydrocarbons as cues for recognizing self and non-self clones. However, the result can be explained if antennae-excised aphids lose the ability to perceive crowding [46].

If aphids can recognize non-clonal members, one might ask whether the members of two different clones would be distributed randomly or distantly from each other on a leaf. Comparisons of the distances between aphid pairs suggested that the distribution of yellow aphids did not completely overlap with that of green aphids in every cage. However, the fact that the percentage of short distances between G–Y pairs was not significantly different from that of Y–Y pairs implies that members of a given clone were not isolated from non-clonal members, such that chemical communication was possible between them.

The present study highlighted, for the first time, the ability of aphids to differentiate between self and non-self clones, and to employ reproductive restraint. Throughout the reproductive season from spring to autumn, aphids of a clone always coexist with different clonal members on host plants [19–25], with pre-empting suitable plant parts and overwhelming the competitors on some plants, while, being forestalled and outnumbered on other plants. Our study indicates that aphids have been selected to evolve the capability of context-dependent decision making such that they accelerate or restrain reproduction to flexibly cope with unexpected competitive situations. In another aphid species, *Myzus persicae*, an increase in colony growth rate was detected under clone–clone competition, and the cause was ascribed to rapid evolutionary changes in clonal composition [26,27]. Although different species may respond differently to competitive situations, it may be important to test whether *M. persicae* has the ability to accelerate or restrain reproduction depending on the relative density of self and non-self clones.

Since our observation was limited to 15 days of aphid reproduction, the outcome of longer-term competition was not elucidated. It has been reported that, in some pairs of aphid clones, the outcome of competition is reversed between low-density and high-density treatments [47]. Thus, more studies are needed to understand the outcome of clone–clone competition when the population density becomes higher or after the population is drastically decreased due to extremely high density and attacks by natural enemies [48]. Nevertheless, long-term studies on clonal competition in aphids, *Daphnia*, and malaria parasites have reported that clones with higher reproductive rates or clones that infested their hosts first had absolute advantages over competitors [25–27,38,49,50]. Therefore, if aphids have the potential to regulate their reproduction, thereby evading hopeless competition, then it could be predicted that clone–clone competition leads to a distinct winner/loser outcome during the initial stage of colony growth. The next step for future studies would be to explore how defeated clones manage to resume reproduction, to what extent the relatedness of coexisting clones is associated with the strength of competition, and to what extent the components of cuticular hydrocarbons are related to self–non-self recognition in aphids.

Data accessibility. The data used in this manuscript are available from the Dryad Digital Repository: https://doi.org/10.5061/dryad. 8gtht76mh [51]. The data are provided in electronic supplementary material [52].

Authors' Contributions. Y.L.: formal analysis, investigation, writing-original draft; S.A.: conceptualization, data curation, formal analysis, funding acquisition, investigation, project administration, visualization, writing-original draft, writing-review & editing. All authors gave final approval for publication and agreed to be held accountable for the work performed therein.

**Competing interests.** We declare we have no competing interests.

**Funding.** This work was supported by Grants-in-Aid (19K06848) for Scientific Research from the Japan Society for the Promotion of Science to S.A.

**Acknowledgements.** We thank Takashi Kanbe for having found the yellow mutant from a number of *A. pisum* clone stocks and providing it for us, Toshiko Taniguchi for helping with cross experiments, and Mayako Kutsukake for helpful suggestions for the experimental design.

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
