## [Peer Review File · Proceedings of the Royal Society B: Biological Sciences]

Review History

RSPB-2020-1856.R0 (Original submission)

Review form: Reviewer 1 (Jean-Christophe Simon)

Recommendation

Major revision is needed (please make suggestions in comments)

Scientific importance: Is the manuscript an original and important contribution to its field?
Excellent

General interest: Is the paper of sufficient general interest?
Excellent

Quality of the paper: Is the overall quality of the paper suitable?
Good

Is the length of the paper justified?
Yes

Should the paper be seen by a specialist statistical reviewer?
No

Do you have any concerns about statistical analyses in this paper? If so, please specify them explicitly in your report.

Yes

It is a condition of publication that authors make their supporting data, code and materials available - either as supplementary material or hosted in an external repository. Please rate, if applicable, the supporting data on the following criteria.

Is it accessible?

Yes

Is it clear?

Yes

Is it adequate?

Yes

Do you have any ethical concerns with this paper?

No

Comments to the Author

This paper presents an elegant study aimed at testing whether aphids can discriminate self from non-self-clones using a green clone and its derived yellow mutant. The experiments are carefully conducted and lead to clear-cut results about the impact of pure vs mixed colonies on clonal competition and reproduction. The paper is well written and contains important results that are relevant for and worth publishing in *Proceedings B*.

However, I have a series of criticisms that require a major revision of the paper. One important point deals with the authors' claim that aphids discriminate self from non-self-clones, which I find not well supported by the results and should be tone-down both in the title and in the discussion. Authors should carefully examine the possibility that their results rather reflect the ability of aphids to perceive variation in density rather than kin recognition. Since the two clones differ in their performances (the green clone being more fecund than the yellow one), this leads to differences in colony density between the yellow and the green clones that might better explain the results than self to non-self recognition. Hereafter are my detailed comments.

L32: replace by "the colony size of the yellow clone was smaller than the one of the green clone". As it is, it sounds you measure the size of the individual aphids (and not the number of offspring in the containers).

L56: replace "during reproduction" by "in the wild"

L66: replace "this feature can be applied" by "this feature applies"

L100: replace "paternity and maternity" by "paternal and maternal origin"

L118-122: explain the purpose of each treatment

L125: If I understood well, the plant leaves were replaced at least 3 times during the course of the experiment. Correct? How did you proceed to change the leaves and did it affect movements in the colonies (therefore spatial interactions among individuals and their distribution)?

L133: I doubt a single excised leaf of *Vicia faba* can hold 130-150 aphids. How many leaves were used per replicate and how aphids were distributed among the leaves? A critical point here is that the focal aphid introduced in the high-density colony would be exposed to spatial differences in density between leaves. How did you control for that? In addition, since the yellow clone is less fecund than the green clone (figure 1a), how did you account for that in the high-density experiment as the number of offspring of the 5 adults would differ de facto between the yellow and the green clones?

L138-141: I understand the design used to discriminate between newborns produced by the transferred aphid from the other ones. However, are you sure there is no risk of confusion, as the oldest aphids from the high-density colonies would be 5 days-old when the 4th instar aphid would be introduced, leaving the possibility that they will start reproducing within the experimental period?

L142-145: this is a weak point of the paper. Why not having introduced an antennae-ablated yellow aphid in a highly dense colony of the yellow clone? With the design used by authors, we cannot conclude from this experiment that aphids detect self from non-self through antennae. The results on ablated vs non-ablated aphids can just reflect the fact that aphids perceive density variation, which is known from earlier works on wing induction in aphids (see for example Kunert & Weisser, *Bull Ent Res* 2005). Similarly, why not having introduced an antennae-ablated green aphid in a highly dense colony of the yellow clone to test whether the supposed discrimination ability also exists in the green clone, giving more support to this hypothesis. The supplementary data on cuticular hydrocarbons cannot be used as evidence that aphids use these cues to discriminate between self and non-self even if some differences between the yellow and green clones were found. Any difference between the two clones can be used as a cue (e.g. color), in absence of more robust evidence that aphids actually discriminate self to non-self and not just crowded vs non-crowded colonies.

L150: Which colonies? You have to make it clear here that you analysed the spatial distribution in the same experiments as described before.

L154-156: Why making two classes of distances? Why not performing the statistical analyses on the absolute distances between each pair of individuals?

L182-184: Since aphid density varies between replicates and treatments, I strongly recommend to introduce aphid density as a covariate in the model.

L203: replace by "after day 9"

L207: replace "much fewer" by "five times less"

L209-210: there is no comparison of the colony size of 1G or 1Y alone vs 1G or 1Y in competition with self or non-self-clones. This is an important result since it allows to quantify the strength of the competition in the different situations and its impact on reproduction (see also comments about the discussion part).

L223 the fecundity reduction is not statistically supported. Please take out this statement.

L229-230: see my previous comment. The results rather suggest that the ablated aphids are not able to perceive crowding.

The discussion is weak and needs a major improvement.

In many instances discussion repeats what is presented in the result section.

The eco-evolutionary consequences of the results are not enough discussed. For example, there is no explanation to why aphids restrain reproduction in self-conditions and not in non-self-conditions. What would be the selective advantage of such a mechanism?

Authors should also provide a hypothesis to the apparently contradictory results obtained in low- and high-density conditions: 1G with 1Y performs better than with 1G in the first experiment (Figure 1c), while it is the reverse when facing high density colonies (Figure 3b). The same applies to the yellow clone.

The authors do not provide strong evidence for an ability to discriminate self to non-self aphids. Please tone down this claim.

L238: replace "elucidate" by "examine"

L243-248: this part should move to the result section and be presented in more details. We don't know for example if the comparisons were made after dividing 2G or 2Y colony size by 2 and whether the differences are statistically significant.

L243 and L246: authors are talking about a density-dependent control of reproduction but I could not find the data which are indicative for that. This comes from the fact that these data are not presented in the result section (see comment L209-210).

L260-261: You cannot conclude from these results that aphids can discriminate between self and non-self clone members.

L271: replace by 'figure'

L479: replace "aphid" by "clone"

Figure 1: Please indicate that the number of aphids is cumulated over the examined reproductive period (15 days)

Figure S4: It would be good to provide the correlation circle of the CHCs

Review form: Reviewer 2

Recommendation

Major revision is needed (please make suggestions in comments)

Scientific importance: Is the manuscript an original and important contribution to its field?

Good

General interest: Is the paper of sufficient general interest?

Good

Quality of the paper: Is the overall quality of the paper suitable?

Good

Is the length of the paper justified?

Yes

Should the paper be seen by a specialist statistical reviewer?

No

Do you have any concerns about statistical analyses in this paper? If so, please specify them explicitly in your report.

No

It is a condition of publication that authors make their supporting data, code and materials available - either as supplementary material or hosted in an external repository. Please rate, if applicable, the supporting data on the following criteria.

Is it accessible?

No

Is it clear?

N/A

Is it adequate?

N/A

Do you have any ethical concerns with this paper?

No

Comments to the Author

This paper provides evidence of self/nonsel self recognition and in aphids. The authors elaborately measure the reproductive rates of different aphid clones by finding a color mutant clone and a new rearing system, which is easy to track each clone separately. The authors found that an aphid clone in a mixed clone colony suppresses reproduction when outnumbered by a different clone. In addition, by cutting the antennae, the authors suggest the aphids use their antennae to recognize self/nonsel self clones. Considering that previous studies in social aphids showed negative results for kin recognition, the findings of self/nonsel self clonal recognition in the yellow and green *A. pisum* clones, which are highly genetically related, would be unexpected and very interesting, and provide a foundation for studying competition and cooperation in mobile organisms.

However, I have some concerns about the interpretation of the data shown in Figure 3, which suggest that aphids recognize self andonsel self clones by using their antennae and suppress reproduction if they are a minor clone.

- The experiment of antennae excision does not have controls, thus cannot rule out the possibility that the antennae excision itself affects the fecundity of aphids as well as self-nonsel self recognition.
- Lines 136-141. It seems that the five adult aphids were not removed from "1Y in a dense G colony" and "1G in a dense Y colony" as done from "1Y in a dense Y colony" and "1G in a dense G colony". Is that right? I think the difference is quite important, because if the five adults remained and continued to reproduce in the colony, the number of opponents rapidly increased, thus the decline in fecundity could be explained just by the difference in the degree of competition between the two experiments.
- "1Y in a dense Y colony" and "1G in a dense G colony" did not show a significant decline compared to control. Figures 1c and 1d showed that the presence of its own clone decreases the fecundity of an aphid. The authors should explain the difference in outcome between the two experiments.
- In general, this experiment cannot distinguish whether the antennae excision loses the ability to recognize the presence of the other clones or to perceive the density of the colony.
- Although not essential, the antennae excision of green-clone aphids would make this study more reliable.

Other comments.

- Please add sample sizes to the figures or figure legends.
- Figure S2 – the information is included in figure 1. Is it necessary?
- Figure 3. I prefer showing dot plots of the data rather than mean±SD.
- Table S3, S4 and figure S4. These results are too immature. PC calculations using the relative proportions of hydrocarbons should be avoided. In addition, the data of other strains or species should be provided to know how much the difference between the two clones is.
- Line 201. figure S1 -> figure S2?
- Line 279. figure S3 -> figure S4?
- Line 222. The use of "Conversely" should be rephrased. The numbers of newborn nymphs in both experiments are smaller than the control. The same holds true for "In contrast" on line 266.

Decision letter (RSPB-2020-1856.R0)

10-Sep-2020

Dear Dr Akimoto:

I am writing to inform you that your manuscript RSPB-2020-1856 entitled "Self and nonself recognition affects clonal reproduction and competition in the pea aphid; withdraw from hopeless competition" has, in its current form, been rejected for publication in Proceedings B.

This action has been taken on the advice of referees, who have recommended that substantial revisions are necessary. With this in mind we would be happy to consider a resubmission, provided the comments of the referees are fully addressed. However please note that this is not a provisional acceptance.

Sincerely,
 Dr Locke Rowe
 mailto: proceedingsb@royalsociety.org

Associate Editor

Comments to Author:

Both referees were enthusiastic about the potential importance of the study and appreciated the creative methodology used to test their hypotheses. Both had some serious concerns, however. Most importantly, both referees took issue with the claim of kin recognition, related to both the experimental methodology and interpretation based on of this part of the study.

The authors need to address Referee 1's requests for clarifications of methods, addressing of statistical concerns, improvements in the discussion (more interpretation of importance in ecological-evolutionary context), and most importantly, the serious concern that they have not demonstrated kin recognition (rather than detection of density).

Referee 2 was similarly concerned with the same experiment, mentioning specifically that the antennal ablation experiment did not have controls and is not correctly interpreted as kin recognition.. The referee also had several additional suggestions to improve the manuscript that need to be addressed..

In addition, I suggest changing the title to something less awkward without a semicolon.

Reviewer(s)' Comments to Author:

Referee: 1

Comments to the Author(s)

This paper presents an elegant study aimed at testing whether aphids can discriminate self from non-self-clones using a green clone and its derived yellow mutant. The experiments are carefully conducted and lead to clear-cut results about the impact of pure vs mixed colonies on clonal competition and reproduction. The paper is well written and contains important results that are relevant for and worth publishing in Proceedings B.

However, I have a series of criticisms that require a major revision of the paper. One important point deals with the authors' claim that aphids discriminate self from non-self-clones, which I find not well supported by the results and should be tone-down both in the title and in the discussion. Authors should carefully examine the possibility that their results rather reflect the ability of aphids to perceive variation in density rather than kin recognition. Since the two clones differ in their performances (the green clone being more fecund than the yellow one), this leads to differences in colony density between the yellow and the green clones that might better explain the results than self to non-self recognition. Hereafter are my detailed comments.

L32: replace by “the colony size of the yellow clone was smaller than the one of the green clone”. As it is, it sounds you measure the size of the individual aphids (and not the number of offspring in the containers).

L56: replace “during reproduction” by “in the wild”

L66: replace “this feature can be applied” by “this feature applies”

L100: replace “paternity and maternity” by “paternal and maternal origin”

L118-122: explain the purpose of each treatment

L125: If I understood well, the plant leaves were replaced at least 3 times during the course of the experiment. Correct? How did you proceed to change the leaves and did it affect movements in the colonies (therefore spatial interactions among individuals and their distribution)?

L133: I doubt a single excised leaf of *Vicia faba* can hold 130-150 aphids. How many leaves were used per replicate and how aphids were distributed among the leaves? A critical point here is that the focal aphid introduced in the high-density colony would be exposed to spatial differences in density between leaves. How did you control for that? In addition, since the yellow clone is less fecund than the green clone (figure 1a), how did you account for that in the high-density experiment as the number of offspring of the 5 adults would differ de facto between the yellow and the green clones?

L138-141: I understand the design used to discriminate between newborns produced by the transferred aphid from the other ones. However, are you sure there is no risk of confusion, as the oldest aphids from the high-density colonies would be 5 days-old when the 4th instar aphid would be introduced, leaving the possibility that they will start reproducing within the experimental period?

L142-145: this is a weak point of the paper. Why not having introduced an antennae-ablated yellow aphid in a highly dense colony of the yellow clone? With the design used by authors, we cannot conclude from this experiment that aphids detect self from non-self through antennae. The results on ablated vs non-ablated aphids can just reflect the fact that aphids perceive density variation, which is known from earlier works on wing induction in aphids (see for example Kunert & Weisser, *Bull Ent Res* 2005). Similarly, why not having introduced an antennae-ablated green aphid in a highly dense colony of the yellow clone to test whether the supposed discrimination ability also exists in the green clone, giving more support to this hypothesis. The supplementary data on cuticular hydrocarbons cannot be used as evidence that aphids use these cues to discriminate between self and non-self even if some differences between the yellow and green clones were found. Any difference between the two clones can be used as a cue (e.g. color), in absence of more robust evidence that aphids actually discriminate self to non-self and not just crowded vs non-crowded colonies.

L150: Which colonies? You have to make it clear here that you analysed the spatial distribution in the same experiments as described before.

L154-156: Why making two classes of distances? Why not performing the statistical analyses on the absolute distances between each pair of individuals?

L182-184: Since aphid density varies between replicates and treatments, I strongly recommend to introduce aphid density as a covariate in the model.

L203: replace by “after day 9”

L207: replace “much fewer” by “five times less”

L209-210: there is no comparison of the colony size of 1G or 1Y alone vs 1G or 1Y in competition with self or non-self-clones. This is an important result since it allows to quantify the strength of the competition in the different situations and its impact on reproduction (see also comments about the discussion part).

L223 the fecundity reduction is not statistically supported. Please take out this statement.

L229-230: see my previous comment. The results rather suggest that the ablated aphids are not able to perceive crowding.

The discussion is weak and needs a major improvement.

In many instances discussion repeats what is presented in the result section.

The eco-evolutionary consequences of the results are not enough discussed. For example, there is no explanation to why aphids restrain reproduction in self-conditions and not in non-self-conditions. What would be the selective advantage of such a mechanism?

Authors should also provide a hypothesis to the apparently contradictory results obtained in low- and high-density conditions: 1G with 1Y performs better than with 1G in the first experiment (Figure 1c), while it is the reverse when facing high density colonies (Figure 3b). The same applies to the yellow clone.

The authors do not provide strong evidence for an ability to discriminate self to non-self aphids. Please tone down this claim.

L238: replace “elucidate” by “examine”

L243-248: this part should move to the result section and be presented in more details. We don't know for example if the comparisons were made after dividing 2G or 2Y colony size by 2 and whether the differences are statistically significant.

L243 and L246: authors are talking about a density-dependent control of reproduction but I could not find the data which are indicative for that. This comes from the fact that these data are not presented in the result section (see comment L209-210).

L260-261: You cannot conclude from these results that aphids can discriminate between self and non-self clone members.

L271: replace by ‘figure’

L479: replace “aphid” by “clone”

Figure 1: Please indicate that the number of aphids is cumulated over the examined reproductive period (15 days)

Figure S4: It would be good to provide the correlation circle of the CHCs

Referee: 2

Comments to the Author(s)

This paper provides evidence of self/nonsel self recognition and in aphids. The authors elaborately measure the reproductive rates of different aphid clones by finding a color mutant clone and a new rearing system, which is easy to track each clone separately. The authors found that an aphid clone in a mixed clone colony suppresses reproduction when outnumbered by a different clone. In addition, by cutting the antennae, the authors suggest the aphids use their antennae to recognize self/nonsel self clones. Considering that previous studies in social aphids showed negative results for kin recognition, the findings of self/nonsel self clonal recognition in the yellow and green *A. pisum* clones, which are highly genetically related, would be unexpected and very interesting, and provide a foundation for studying competition and cooperation in mobile organisms.

However, I have some concerns about the interpretation of the data shown in Figure 3, which suggest that aphids recognize self andonsel self clones by using their antennae and suppress reproduction if they are a minor clone.

- The experiment of antennae excision does not have controls, thus cannot rule out the possibility that the antennae excision itself affects the fecundity of aphids as well as self-nonsel self recognition.

- Lines 136-141. It seems that the five adult aphids were not removed from “1Y in a dense G colony” and “1G in a dense Y colony” as done from “1Y in a dense Y colony” and “1G in a dense G colony”. Is that right? I think the difference is quite important, because if the five adults remained and continued to reproduce in the colony, the number of opponents rapidly increased, thus the decline in fecundity could be explained just by the difference in the degree of competition between the two experiments.

- “1Y in a dense Y colony” and “1G in a dense G colony” did not show a significant decline compared to control. Figures 1c and 1d showed that the presence of its own clone decreases the fecundity of an aphid. The authors should explain the difference in outcome between the two experiments.

- In general, this experiment cannot distinguish whether the antennae excision loses the ability to recognize the presence of the other clones or to perceive the density of the colony.

- Although not essential, the antennae excision of green-clone aphids would make this study more reliable.

Other comments.

- Please add sample sizes to the figures or figure legends.
- Figure S2 – the information is included in figure 1. Is it necessary?
- Figure 3. I prefer showing dot plots of the data rather than mean \pm SD.
- Table S3, S4 and figure S4. These results are too immature. PC calculations using the relative proportions of hydrocarbons should be avoided. In addition, the data of other strains or species should be provided to know how much the difference between the two clones is.
- Line 201. figure S1 -> figure S2?
- Line 279. figure S3 -> figure S4?
- Line 222. The use of “Conversely” should be rephrased. The numbers of newborn nymphs in both experiments are smaller than the control. The same holds true for “In contrast” on line 266.

Author's Response to Decision Letter for (RSPB-2020-1856.R0)

See Appendix A.

RSPB-2021-0787.R0

Review form: Reviewer 1

Recommendation

Accept with minor revision (please list in comments)

Scientific importance: Is the manuscript an original and important contribution to its field?

Good

General interest: Is the paper of sufficient general interest?

Good

Quality of the paper: Is the overall quality of the paper suitable?

Good

Is the length of the paper justified?

Yes

Should the paper be seen by a specialist statistical reviewer?

No

Do you have any concerns about statistical analyses in this paper? If so, please specify them explicitly in your report.

Yes

It is a condition of publication that authors make their supporting data, code and materials available - either as supplementary material or hosted in an external repository. Please rate, if applicable, the supporting data on the following criteria.

Is it accessible?

No

Is it clear?

N/A

Is it adequate?

N/A

Do you have any ethical concerns with this paper?

No

Comments to the Author

Review of revised version of Li et al. paper

I would like first to thank the authors for their extensive revision of the previous version of the paper and for adding new results that consolidate the hypothesis for recognition of self and non-self in aphids. However, I still have some comments that authors should consider to clarify some parts of the MS and more carefully interpret and discuss their results.

- 1) You should better explain from line 125 why you performed an experiment where you introduce the green clone several days after the yellow one started to reproduce. Explain what is the rationale behind. Related to this experiment, I don't understand the figure 3. Why do you have negative numbers on the X axis? Why not simply give the number of yellow nymphs at the time you introduced the green clone. This is very confusing. Did you pool the results of two experiments (the one without time difference in the installation of the two clones and the one with)? Why did you draw a line to relate the datapoints? I doubt there is a linear relationship between pre-existing yellow nymphs and % of yellow aphids. It looks more there is a threshold function, with proportion of yellow aphids changing dramatically from 5 to 10 yellow nymphs. You should give more details in material and methods on how you analyse the data of this experiment. We discovered from lines 237 to 247 you performed a logistic regression and other stats (AIC) that are not presented. "Than when it was highly biased" is awkward. I guess you want to say "than for smaller or larger values". You also need to say briefly after line 247 what you conclude from these results.
- 2) For the experiment using high density colonies, could you give the age range of the 130-150 aphids at the time you introduced the focal yellow aphid. Could you also give the age of the five aphids that are let to reproduce for 5 days. Did you remove all adults for both conditions (when 1 Y is introduced in high density Y and 1 Y in high density G) and at the same time? Please clarify.
- 3) Since you did not perform the reciprocal experiments (high density and antennae ablation) for the green clone, the result that aphids recognize self from non-self only stands for the yellow clone. This should be acknowledged in the discussion. Although I disagree with lines 270-272 since the green clone doesn't restrain reproduction in presence of the yellow one, at least not in the second reproductive period (from day 10). I really want the authors avoid generalizing their results as what is valid for the yellow clone is not for the green one, or has not been tested. It may be good in the discussion to compare the "strategies" of the two clones, which may rely on different cues and mechanisms.
- 4) Lines 212-213, test if significantly different and give p value.
- 5) Line 290-293: I don't understand what you want to say and which experiments you are referring to.
- 6) Line 304: which transfer experiment? Are you talking about the introduction of single aphids into high density colonies?
- 7) Line 314: "might be advantageous" and not "is advantageous" otherwise it's very speculative.
- 8) Lines 318-320, since you removed the part on cuticular hydrocarbons, we have no evidence that the two clones differ in chemical substances. Please rephrase.
- 9) Lines 342-344, there are multiple examples of clonal mixing (different genotypes) on the same plant/colonies. Please rephrase.

Review form: Reviewer 2

Recommendation

Accept with minor revision (please list in comments)

Scientific importance: Is the manuscript an original and important contribution to its field?

Excellent

General interest: Is the paper of sufficient general interest?

Good

Quality of the paper: Is the overall quality of the paper suitable?

Good

Is the length of the paper justified?

Yes

Should the paper be seen by a specialist statistical reviewer?

No

Do you have any concerns about statistical analyses in this paper? If so, please specify them explicitly in your report.

No

It is a condition of publication that authors make their supporting data, code and materials available - either as supplementary material or hosted in an external repository. Please rate, if applicable, the supporting data on the following criteria.

Is it accessible?

No

Is it clear?

N/A

Is it adequate?

N/A

Do you have any ethical concerns with this paper?

No

Comments to the Author

The authors addressed my concern about the antennal ablation experiment by adding controls of the antennae ablation experiments. In addition, the authors conducted a new growth competition experiment and found that yellow aphids with a lower reproductive rate can overwhelm green aphids in number when they start to reproduce a little earlier. They also improve the statistical analyses.

I cannot understand what the authors show in the new Fig. 3a. What does the strange minus number mean? Does it mean the authors also let the green clone start reproduction earlier than the yellow one? However, there is no remark in the materials and methods. Authors should indicate the number of the yellow clones produced during the first two or three days in the results, and also describe in the figure legends what the plotted samples represent.

Please indicate that the antennae-ablated aphids might not lose the ability to recognize self/nonsel but to perceive crowding. They mention it very briefly in the text (Lines 253-254 and

331-332), but I do not think it is enough. This result cannot rule out the possibility that the antennae just perceive crowding, as the authors admitted in their Response. They should tone down their statements in the abstract and the discussion, which can be misleading.

Other comments.

- Please add sample sizes to Figures 1-2 or their legends.
- Lines 49 and 61. kin and non-kin clone -> closely related and unrelated clone?

Decision letter (RSPB-2021-0787.R0)

07-May-2021

Dear Dr Akimoto

I am pleased to inform you that your manuscript RSPB-2021-0787 entitled "Self and nonself recognition affects clonal reproduction and competition in the pea aphid" has been accepted for publication in Proceedings B.

The referee(s) have recommended publication, but also suggest some minor revisions to your manuscript. Therefore, I invite you to respond to the referee(s)' comments and revise your manuscript. Because the schedule for publication is very tight, it is a condition of publication that you submit the revised version of your manuscript within 7 days. If you do not think you will be able to meet this date please let us know.

Sincerely,

Dr Locke Rowe

Associate Editor

Board Member

Comments to Author:

The referees appreciated the care and effort given to the revisions, and continue to recognize the importance of this study. At this stage, the reviewers still find some additional areas of the paper that need attention and revision, I recommend these be required before the paper can be accepted. These include revisions to Figure 3a, inclusion of sample sizes in Figures 1-2, several text clarifications, and some toning down of the interpretation so as to not over-generalize the implications of the results. I suggest the authors pay careful attention to these suggestions and

address them fully. Another round of revisions will not be permitted, and some of these are important concerns and therefore it is imperative they are addressed thoroughly.

Reviewer(s)' Comments to Author:

Referee: 2

Comments to the Author(s).

The authors addressed my concern about the antennal ablation experiment by adding controls of the antennae ablation experiments. In addition, the authors conducted a new growth competition experiment and found that yellow aphids with a lower reproductive rate can overwhelm green aphids in number when they start to reproduce a little earlier. They also improve the statistical analyses.

I cannot understand what the authors show in the new Fig. 3a. What does the strange minus number mean? Does it mean the authors also let the green clone start reproduction earlier than the yellow one? However, there is no remark in the materials and methods. Authors should indicate the number of the yellow clones produced during the first two or three days in the results, and also describe in the figure legends what the plotted samples represent.

Please indicate that the antennae-ablated aphids might not lose the ability to recognize self/nonsel but to perceive crowding. They mention it very briefly in the text (Lines 253-254 and 331-332), but I do not think it is enough. This result cannot rule out the possibility that the antennae just perceive crowding, as the authors admitted in their Response. They should tone down their statements in the abstract and the discussion, which can be misleading.

Other comments.

- Please add sample sizes to Figures 1-2 or their legends.
- Lines 49 and 61. kin and non-kin clone -> closely related and unrelated clone?

Referee: 1

Comments to the Author(s).

Review of revised version of Li et al. paper

I would like first to thank the authors for their extensive revision of the previous version of the paper and for adding new results that consolidate the hypothesis for recognition of self and non-self in aphids. However, I still have some comments that authors should consider to clarify some parts of the MS and more carefully interpret and discuss their results.

- 1) You should better explain from line 125 why you performed an experiment where you introduce the green clone several days after the yellow one started to reproduce. Explain what is the rationale behind. Related to this experiment, I don't understand the figure 3. Why do you have negative numbers on the X axis? Why not simply give the number of yellow nymphs at the time you introduced the green clone. This is very confusing. Did you pool the results of two experiments (the one without time difference in the installation of the two clones and the one with)? Why did you draw a line to relate the datapoints? I doubt there is a linear relationship between pre-existing yellow nymphs and % of yellow aphids. It looks more there is a threshold function, with proportion of yellow aphids changing dramatically from 5 to 10 yellow nymphs. You should give more details in material and methods on how you analyse the data of this experiment. We discovered from lines 237 to 247 you performed a logistic regression and other stats (AIC) that are not presented. "Than when it was highly biased" is awkward. I guess you want to say "than for smaller or larger values". You also need to say briefly after line 247 what you conclude from these results.
- 2) For the experiment using high density colonies, could you give the age range of the 130-150 aphids at the time you introduced the focal yellow aphid. Could you also give the age of the five aphids that are let to reproduce for 5 days. Did you remove all adults for both conditions (when 1 Y is introduced in high density Y and 1 Y in high density G) and at the same time? Please clarify.

- 3) Since you did not perform the reciprocal experiments (high density and antennae ablation) for the green clone, the result that aphids recognize self from non-self only stands for the yellow clone. This should be acknowledged in the discussion. Although I disagree with lines 270-272 since the green clone doesn't restrain reproduction in presence of the yellow one, at least not in the second reproductive period (from day 10). I really want the authors avoid generalizing their results as what is valid for the yellow clone is not for the green one, or has not been tested. It may be good in the discussion to compare the "strategies" of the two clones, which may rely on different cues and mechanisms.
- 4) Lines 212-213, test if significantly different and give p value.
- 5) Line 290-293: I don't understand what you want to say and which experiments you are referring to.
- 6) Line 304: which transfer experiment? Are you talking about the introduction of single aphids into high density colonies?
- 7) Line 314: "might be advantageous" and not "is advantageous" otherwise it's very speculative.
- 8) Lines 318-320, since you removed the part on cuticular hydrocarbons, we have no evidence that the two clones differ in chemical substances. Please rephrase.
- 9) Lines 342-344, there are multiple examples of clonal mixing (different genotypes) on the same plant/colonies. Please rephrase.

Author's Response to Decision Letter for (RSPB-2021-0787.R0)

See Appendix B.

Decision letter (RSPB-2021-0787.R1)

27-May-2021

Dear Dr Akimoto

I am pleased to inform you that your manuscript RSPB-2021-0787.R1 entitled "Self and nonself recognition affects clonal reproduction and competition in the pea aphid" has been accepted for publication in Proceedings B.

The board member has recommended publication, but also suggest some minor revisions to your manuscript. Therefore, I invite you to respond to their comments and revise your manuscript. Because the schedule for publication is very tight, it is a condition of publication that you submit the revised version of your manuscript within 7 days. If you do not think you will be able to meet this date please let us know.

When submitting your revised manuscript, you will be able to respond to the comments made by the referee(s) and upload a file "Response to Referees". You can use this to document any changes you make to the original manuscript. We require a copy of the manuscript with revisions made

since the previous version marked as 'tracked changes' to be included in the 'response to referees' document.

Sincerely,
Dr Locke Rowe
Editor, Proceedings B
<mailto:proceedingsb@royalsociety.org>

Associate Editor:

Board Member: 1

Comments to Author:

The authors have done a fine job on the revisions, but I found multiple grammatical errors (highlighted in yellow in the word doc attached) and one sentence that is confusing. The authors also did not add sample sizes to Figures 1 and 2 as requested. Please correct these in the next draft, another round of revisions will not be allowed. -- Amy Toth, Board Member

Decision letter (RSPB-2021-0787.R2)

04-Jun-2021

Dear Dr Akimoto

I am pleased to inform you that your manuscript entitled "Self and nonself recognition affects clonal reproduction and competition in the pea aphid" has been accepted for publication in Proceedings B.

Your article has been estimated as being 9 pages long. Our Production Office will be able to confirm the exact length at proof stage.

Data Accessibility section

Open Access

You are invited to opt for Open Access, making your freely available to all as soon as it is ready for publication under a CCBY licence. Our article processing charge for Open Access is £1700. Corresponding authors from member institutions

Paper charges

Sincerely,

Appendix A

Associate Editor

Comments to Author:

Both referees were enthusiastic about the potential importance of the study and appreciated the creative methodology used to test their hypotheses. Both had some serious concerns, however. Most importantly, both referees took issue with the claim of kin recognition, related to both the experimental methodology and interpretation based on this part of the study.

The authors need to address Referee 1's requests for clarifications of methods, addressing of statistical concerns, improvements in the discussion (more interpretation of importance in ecological-evolutionary context), and most importantly, the serious concern that they have not demonstrated kin recognition (rather than detection of density).

Referee 2 was similarly concerned with the same experiment, mentioning specifically that the antennal ablation experiment did not have controls and is not correctly interpreted as kin recognition.. The referee also had several additional suggestions to improve the manuscript that need to be addressed..

In addition, I suggest changing the title to something less awkward without a semicolon.

Response: Thank you very much for permitting us to revise the manuscript, and we are very sorry for the large delay in returning the revised manuscript. During this period, we conducted new experiments, which are requested by the reviewers, and other original experiments. Based on the results, we have revised most parts of the text, figures and tables, and deleted some electronic supplementary materials following the suggestions by the reviewers. In this revision, we have revised all the sentences on which Reviewer 1 and 2 had concerns, and changed the title following the suggestion by the associate editor.

In the following, we responded to all the comments by the reviewers sequentially.

We are very happy if this revision and responses relieve the concerns posed by the reviewers.

Reviewer(s)' Comments to Author:

Referee: 1

Comments to the Author(s)

This paper presents an elegant study aimed at testing whether aphids can discriminate self from non-self-clones using a green clone and its derived yellow mutant. The experiments are carefully conducted and lead to clear-cut results about the impact of pure vs mixed colonies on clonal competition and reproduction. The paper is well written and contains important results that are relevant for and worth publishing in Proceedings B.

However, I have a series of criticisms that require a major revision of the paper. One important point deals with the authors' claim that aphids discriminate self from non-self-clones, which I find not well supported by the results and should be tone-down both in the title and in the discussion. Authors should carefully examine the possibility that their results rather reflect the ability of aphids to perceive variation in density rather than kin recognition. Since the two clones differ in their performances (the green clone being more fecund than the yellow one), this leads to differences in colony density between the yellow and the green clones that might better explain the results than self to non-self recognition. Hereafter are my detailed comments.

Response: Thank you very much for the valuable comments, and we are very sorry for the delay in returning the revised manuscript. We have conducted new experiments as well as additional experiments following the reviewers' suggestions and have revised the manuscript based on the results. In this revision, we followed all the comments by Reviewer 1 and 2.

In the previous version of MS, we observed competition between the green aphids and the yellow aphids and reported that green aphids overwhelm yellow aphids in number because of their higher reproductive rate. This time, we found the condition where yellow aphids outnumber green aphids. That is, if yellow adults start to reproduce a little earlier than do green adults, yellow aphids overwhelm green aphids in number, and green aphids restrain reproduction. For antennae-ablation experiments, we tried to increase replication and conducted a new type of experiments that was lacking in the previous version. Based on the result from new experiments, we would like to emphasize

that aphids can conduct context-dependent decision making such that they accelerate or restrain reproduction according to the relative density of self and non-self clones.

Our claim that aphids discriminate self from non-self-clones comes from (1) different colony growth patterns between mixed colonies (1Y+1G) and clonal colonies (2Y or 2G), and (2) transfer experiments of one yellow adult (1Y) to a dense G vs. Y colony (this version of MS). We think that the difference in the cumulative number between mixed colonies and clonal colonies suggests the presence of competitiveness of both clones, which is based on the recognition of self/non-self clones. In particular, in mixed colonies, although yellow aphids were outnumbered by green aphids at last, 1Y in the 1Y+1G colonies produced nymphs more rapidly than did 1Y in the 2Y colonies by the time when the offspring generation started to reproduce, with the reproductive rate similar to 1Y alone. We think that this is evidence for an aphid regulating reproduction in competitive situations. Transfer of 1Y to a dense G colony vs. a dense Y colony resulted in much reduced fecundity of 1Y in a dense G colony, and we think that this is also evidence for the ability of aphids to discriminate self from non-self clones.

We have revised most parts of the text, figures and tables, and deleted some electronic supplementary materials following the comments by the reviewers.

We are very happy if this revision is satisfactory to the reviewers and is appropriate for publication.

L32: replace by “the colony size of the yellow clone was smaller than the one of the green clone”. As it is, it sounds you measure the size of the individual aphids (and not the number of offspring in the containers).

Response: Thank you for the suggestion. We revised this sentence differently, deleting the phrase.

L56: replace “during reproduction” by “in the wild”

Response: Thank you. We revised the sentence following the suggestion.

L66: replace “this feature can be applied” by “this feature applies”

Response: We revised the sentence following the suggestion.

L100: replace “paternity and maternity” by “paternal and maternal origin”

Response: We revised the sentence following the suggestion.

L118-122: explain the purpose of each treatment

Response: We added a short phrase representing the purpose of each treatment.

L125: If I understood well, the plant leaves were replaced at least 3 times during the course of the experiment. Correct? How did you proceed to change the leaves and did it affect movements in the colonies (therefore spatial interactions among individuals and their distribution)?

Response: In this experiment, we added one new leaf to empty space such that the leaf touch with the old one. Yes, new leaves were added 3 times, but old leaves were not removed. As shown in supplementary figure S2 (in this version), we finally placed four to six leaves (depending on the size) on the agar medium. At each time, a new leaf was placed on empty space, so disturbance was kept minimum. Aphids themselves moved from old and crowded leaves to a new leaf. We did not manipulate or transfer aphids by using a brush. Supplementary figure S2 shows old and new and leaves, which reflect the order of transplant.

L133: I doubt a single excised leaf of *Vicia faba* can hold 130-150 aphids. How many leaves were used per replicate and how aphids were distributed among the leaves? A critical point here is that the focal aphid introduced in the high-density colony would be exposed to spatial differences in density between leaves. How did you control for that? In addition, since the yellow clone is less fecund than the green clone (figure 1a), how did you account for that in the high-density experiment as the number of offspring of the 5 adults would differ de facto between the yellow and the green clones?

Response: We are sorry for the insufficient explanation. We first placed one cut leaf on the agar surface and transferred five adults onto them. Three days after the transplant, one new leaf was added (which was added to Material and Methods). Five days later, there was a highly dense colony on every leaf. A test aphid was introduced on one of the leaves, and then that aphid selected one leaf. We were not able to control where the aphid settled on, but the aphid selected the place. We think that the aphid experienced high density on any leaf. In our observation, the size of the yellow and green clone equally increased up to 130 to 150 because of density-dependent effects. Before transfer,

we confirmed that there were 130 to 150 aphids on the leaves. Although the yellow and green clone differed in initial growth rate, aphids of both clones produced few nymphs or stopped reproduction after the density reached to 130 to 150 aphids.

L138-141: I understand the design used to discriminate between newborns produced by the transferred aphid from the other ones. However, are you sure there is no risk of confusion, as the oldest aphids from the high-density colonies would be 5 days-old when the 4th instar aphid would be introduced, leaving the possibility that they will start reproducing within the experimental period?

Response: We have sufficient information about the age at which aphids start reproduction, and based on the information we made the experimental design. Our records showed that in this system, 10-days old aphids after birth start to reproduce in both clones. There are no exceptions. When we introduced a test adult into a colony, the colony contains 5-days old and younger nymphs. We observed larviposition by the test adult for three days, and at the end of the observation, other aphids (except the test adult) were 8-days old or younger. This means that other aphids did not reach age sufficient for reproduction. We also counted the number of aphids every day (based on photographs) to confirm whether the newborns were produced by the test adult.

L142-145: this is a weak point of the paper. Why not having introduced an antennae-ablated yellow aphid in a highly dense colony of the yellow clone? With the design used by authors, we cannot conclude from this experiment that aphids detect self from non-self through antennae. The results on ablated vs non-ablated aphids can just reflect the fact that aphids perceive density variation, which is known from earlier works on wing induction in aphids (see for example Kunert & Weisser, *Bull Ent Res* 2005). Similarly, why not having introduced an antennae-ablated green aphid in a highly dense colony of the yellow clone to test whether the supposed discrimination ability also exists in the green clone, giving more support to this hypothesis. The supplementary data on cuticular hydrocarbons cannot be used as evidence that aphids use these cues to discriminate between self and non-self even if some differences between the yellow and green clones were found. Any difference between the two clones can be used as a cue (e.g. color), in absence of more robust evidence that aphids actually discriminate self to non-self and not just crowded vs non-crowded colonies.

Response: Thank you very much for the important comments. This is a flaw in our experimental design. Following the comments, we newly conducted experiments in which one antennae-ablated yellow aphid was transferred to a crowded colony of the yellow clone. This experiment was lacking in the previous version. The results are represented in new figure 4. This time, we additionally increased replication for each kind of experiments, with 10 to 13 replicates. The results showed that if intact yellow aphids were transferred to crowded green colonies, they produced much fewer nymphs than did those that were transferred to crowded yellow colonies, whereas antennae-ablated yellow aphids produced as many nymphs as did control aphids, irrespective of whether they were transferred to crowded green or yellow colonies. As suggested in Kunert & Weisser (2005), we also think that aphids use their antennae for recognizing members of a distinct clone and estimating the density. For green aphids, we understand that the same set of experiments are needed. However, we are very sorry that we were not able to complete the same set of experiments for green clones because of out of time. Thus, we decided to remove the incomplete result using green aphids from figures.

We newly cited Kunert & Weisser (2005).

L150: Which colonies? You have to make it clear here that you analysed the spatial distribution in the same experiments as described before.

Response: We are sorry for insufficient explanation. In this respect, we refer to “mixed colonies”, and corrected this point.

L154-156: Why making two classes of distances? Why not performing the statistical analyses on the absolute distances between each pair of individuals?

Response: Thank you very much for the suggestion. Of course, we first conducted ANOVA based on the absolute distances between aphids (green-green members, yellow-yellow members, and green-yellow members). We added the result to the table in this revision. We understand that the mean distance of green-yellow members is larger than the mean distance between aphids of the same clone in every container. This suggests that distribution of a yellow aphid colony and that of a green aphid colony is not completely mixed in a container (difference in the centroid). However, the difference in the mean distance between combinations are very small, with large variance (SD).

Because of large sample size (n is some thousands), the differences easily become significant.

We think that the absolute distances among aphids are not so useful for detecting interactions between clones. This is because aphids are distributed on leaves we set, so absolute distances among aphids are affected by the positions of the leaves. Aphids are not affected chemically by aphids on different leaves, so we expect that interactions between different clones, if present, would emerge among aphids living on the same leaf and in the close vicinity. Thus, we calculated the proportions of aphid pairs (green-green, yellow-yellow, and green-yellow) situated close to each other and tested if the proportion of green-yellow pairs in vicinity is smaller than that of green-green pairs or yellow-yellow pairs in each container. This analysis told us that the proportion of green-yellow pairs in vicinity is similar to that of yellow-yellow pairs, suggesting that green and yellow aphids do not avoid the other clone members.

L182-184: Since aphid density varies between replicates and treatments, I strongly recommend to introduce aphid density as a covariate in the model.

Response: Thank you very much for the helpful suggestion. We introduced aphid density in the model as a covariate, and revised the Results accordingly.

L203: replace by “after day 9”

Response: We revised the sentence following the suggestion.

L207: replace “much fewer” by “five times less”

Response: Thank you. We revised the sentence following the suggestion.

L209-210: there is no comparison of the colony size of 1G or 1Y alone vs 1G or 1Y in competition with self or non-self-clones. This is an important result since it allows to quantify the strength of the competition in the different situations and its impact on reproduction (see also comments about the discussion part).

Response: We newly added the results of statistical comparisons of population growth between 1G alone and 1G in 2G colonies or 1G in 1Y+1G colonies, and between 1Y alone and 1Y in 2Y colonies or 1Y in 1Y+1G colonies. In any case, colonies founded by 1G or 1Y alone, respectively, increased in number more rapidly than 1G or 1Y in two-

aphid colonies (clonal or non-clonal members).

L223 the fecundity reduction is not statistically supported. Please take out this statement.

Response: Thank you for the suggestion. We revised the original sentence to “its fecundity did not change compared to that of 1Y.”

L229-230: see my previous comment. The results rather suggest that the ablated aphids are not able to perceive crowding.

Response: Thank you for the comment. We think that there are three possibilities; ablated aphids are not able to perceive crowding or the presence of other clones, or both. We revised the sentence as follows; “, suggesting that the aphid was not able to perceive the presence or crowding of the different clones.”

The discussion is weak and needs a major improvement.

In many instances discussion repeats what is presented in the result section.

The eco-evolutionary consequences of the results are not enough discussed. For example, there is no explanation to why aphids restrain reproduction in self-conditions and not in non-self-conditions. What would be the selective advantage of such a mechanism?

Authors should also provide a hypothesis to the apparently contradictory results obtained in low- and high-density conditions: 1G with 1Y performs better than with 1G in the first experiment (Figure 1c), while it is the reverse when facing high density colonies (Figure 3b). The same applies to the yellow clone.

The authors do not provide strong evidence for an ability to discriminate self to non-self aphids. Please tone down this claim.

Response: We largely revised the Discussion part, taking account of the results of new experiments, which indicate that yellow aphids with a lower reproductive rate can overwhelm green aphids in number when they start to reproduce a little earlier. This time we emphasized the ability of context-dependent decision making in aphids and that they can accelerate or restrain reproduction to flexibly cope with unexpected competitive situations. We discussed evolutionary consequences of the flexible ability of aphids.

We think that the query of “there is no explanation to why aphids restrain reproduction in self-conditions and not in non-self-conditions. What would be the

selective advantage of such a mechanism?” raises a critical issue (thank you!). Although we have no data at present, we think that the viability and performance of nymphs may differ between the situations where they are surrounded by fellow aphids and where they are surrounded by members of a different clone. In the case of self-crowding conditions, nymphs and other aphids can survive by sharing resources and by becoming smaller adults. Because they are clonal, all aphids would become altruistic to each other. However, in the case of nonself-crowding conditions, different clones probably behave selfishly such that sharing of food resources may be limited. This condition would increase the mortality of nymphs, so that aphids refrain from producing nymphs when they are surrounded by aphids of different clones. That is, the difference in reproductive patterns may be related to the relatedness of clones. In a future study, we would like to test this idea. However, in our case, the yellow and green clones are actually closely related, but because of mutation effects on phenotypes, they seem to recognize each other as a distinct clone. We added short explanation to Discussion.

The second query of “apparently contradictory results obtained in low- and high-density conditions: 1G with 1Y performs better than with 1G in the first experiment (Figure 1c), while it is the reverse when facing high density colonies” is, we think, related to the ability of aphids to regulate their reproduction, depending on whether they are superior or inferior in number to their competitors. For example, when yellow aphids start to reproduce earlier than do green aphids, they can be superior in number to green aphids in the initial stage, and in this condition yellow aphids accelerate their reproduction. However, when yellow aphids start to reproduce simultaneously with green aphids, they will be inferior in number to green aphids in the initial stage because of lower reproductive rate. In this condition, yellow aphids restrain their reproduction. Thus, we think that aphids can conduct context-dependent decision making. We also think that this ability is based on the ability to discriminate different clones.

L238: replace “elucidate” by “examine”

Response: We revised the sentence following the suggestion.

L243-248: this part should move to the result section and be presented in more details. We don't know for example if the comparisons were made after dividing 2G or 2Y colony size by 2 and whether the differences are statistically significant.

Response: We moved this part to Results, in which statistical tests were conducted.

L243 and L246: authors are talking about a density-dependent control of reproduction but I could not find the data which are indicative for that. This comes from the fact that these data are not presented in the result section (see comment L209-210).

Response: Thank you for the comments. We added the results concerning a density-dependent control of reproduction in clonal colonies to the Results part. In addition, the results of statistical tests were added in Results.

L260-261: You cannot conclude from these results that aphids can discriminate between self and non-self clone members.

Response: We weakened the expression and revised this sentence to “These results indicate that aphid clones can regulate their reproductive rate in competitive situations.”

L271: replace by ‘figure’

Response: We revised the sentence following the suggestion.

L479: replace “aphid” by “clone”

Response: We revised the sentence following the suggestion. Certainly, reproduction is not by one aphid.

Figure 1: Please indicate that the number of aphids is cumulated over the examined reproductive period (15 days)

Response: We added the explanation.

Figure S4: It would be good to provide the correlation circle of the CHCs

Response: For CHCs analysis, we deleted all results (electronic supplementary material Table S3, Table S4, and figure S3) following Reviewer 2's comment.

Referee: 2

Comments to the Author(s)

This paper provides evidence of self/nonself recognition and in aphids. The authors elaborately measure the reproductive rates of different aphid clones by finding a color mutant clone and a new rearing system, which is easy to track each clone separately. The authors found that an aphid clone in a mixed clone colony suppresses reproduction when outnumbered by a different clone. In addition, by cutting the antennae, the authors suggest the aphids use their antennae to recognize self/nonself clones. Considering that previous studies in social aphids showed negative results for kin recognition, the findings of self/nonself clonal recognition in the yellow and green *A. pisum* clones, which are highly genetically related, would be unexpected and very interesting, and provide a foundation for studying competition and cooperation in mobile organisms.

Response: Thank you very much for the valuable comments, and we are very sorry for the delay in returning the revised manuscript. We have conducted completely new experiments as well as additional experiments following the reviewers' suggestions and have revised the manuscript based on the results. In the revision, we followed all the comments by the reviewers.

In the previous version of MS, we observed competition between the green aphids and the yellow aphids and reported that green aphids overwhelm yellow aphids in number because of their higher reproductive rate. This time, we found the condition where yellow aphids outnumber green aphids. That is, if yellow adults start to reproduce a little earlier than do green adults, yellow aphids overwhelm green aphids in number, and green aphids restrain reproduction. For antennae-ablation experiments, we tried to increase replication and conducted a new type of experiments that was lacking in the previous version. Based on the result from new experiments, we would like to emphasize that aphids can conduct context-dependent decision making such that they accelerate or restrain reproduction to flexibly cope with unexpected competitive situations.

We have revised most parts of the text, figures and tables, and deleted some electronic supplementary materials following the comments by the reviewers. We are very happy if this revision is satisfactory to the reviewers and is appropriate for publication.

However, I have some concerns about the interpretation of the data shown in Figure 3, which suggest that aphids recognize self and nonself clones by using their antennae and

suppress reproduction if they are a minor clone.

- The experiment of antennae excision does not have controls, thus cannot rule out the possibility that the antennae excision itself affects the fecundity of aphids as well as self-nonsel self recognition.

Response: Thank you very much for the comment. This is a flaw in our experimental design. Following the comment, we newly tried to conduct the control experiment (transfer of an antennae-excised yellow aphid to a dens colony of the self clone). The result shows that the fecundity of antennae-excised yellow aphids did not differ significantly from that of intact yellow aphids in crowded yellow colonies, so we think that antennae excision had negligible effects on their reproduction.

This time, we concentrate on the transfer of one yellow adult to a dense colony, increasing replications. Because of lack of time, we decided to remove the incomplete results of the transfer of green aphids from the figure. We understand that the same set of experiments are needed for green aphids, but we are very sorry that time was not enough to complete the experiment.

- Lines 136-141. It seems that the five adult aphids were not removed from “1Y in a dense G colony” and “1G in a dense Y colony” as done from “1Y in a dense Y colony” and “1G in a dense G colony”. Is that right? I think the difference is quite important, because if the five adults remained and continued to reproduce in the colony, the number of opponents rapidly increased, thus the decline in fecundity could be explained just by the difference in the degree of competition between the two experiments.

Response: Yes, we did not remove five adults from “1Y in a dense G colony” and “1G in a dense Y colony”. The five adults remained in the colony with different clones. However, we observed that when the colony size reached the level of 130 to 150 aphids, the adults almost stopped reproduction because of density-dependent regulation, so we think that the colony size remained at almost the same level for three days and the effect of five adults is negligible. At least, we kept equal conditions between “1 natal Y in a dense G colony” and “1 antennae-excised Y in a dense G colony” or between “1 natal Y in a dense Y colony”, and “1 antennae-excised Y in a dense Y colony”.

- “1Y in a dense Y colony” and “1G in a dense G colony” did not show a significant decline compared to control. Figures 1c and 1d showed that the presence of its own clone

decreases the fecundity of an aphid. The authors should explain the difference in outcome between the two experiments.

Response: Before experiments, we expected that the fecundity of 1Y transferred to a dense Y colony should be smaller than 1Y alone (due to density dependent effects) but larger than the fecundity of 1Y transferred to a dense G colony. This time, we increased replication for the experiment “1Y in a dense Y colony”. However, we were not able to observe a significant decrease in 1Y fecundity compared with the control (1Y alone). This result is difficult to interpret, and we think that one (and only) reason is a small size of replication (n=13, and for other experiments n=10 or 11). This experiment needs delicate settings, so it was difficult to increase replication.

- In general, this experiment cannot distinguish whether the antennae excision loses the ability to recognize the presence of the other clones or to perceive the density of the colony.

- Although not essential, the antennae excision of green-clone aphids would make this study more reliable.

Response: As the reviewer pointed out, antennae-excision experiments cannot distinguish whether antennae are used for recognizing the presence of other clones or perceiving the density. We think that antennae basically contribute to both functions.

Other comments.

- Please add sample sizes to the figures or figure legends.

Response: We added sample size in the figure.

- Figure S2 – the information is included in figure 1. Is it necessary?

Response: Thank you for the suggestion. We deleted the figure.

- Figure 3. I prefer showing dot plots of the data rather than mean±SD.

Response: Thank you for the suggestion. In this revision, we used dot plots plus mean±SE.

- Table S3, S4 and figure S4. These results are too immature. PC calculations using the relative proportions of hydrocarbons should be avoided. In addition, the data of other

strains or species should be provided to know how much the difference between the two clones is.

Response: Thank you for the suggestion. We agree the comment, so we deleted these data (Table S3, S4 and figure S4) concerning to hydrocarbons and the PC.

- Line 201. figure S1 -> figure S2?

Response: sorry, we made a mistake. we changed it to figure 1 c and d.

- Line 279. figure S3 -> figure S4?

Response: Thank you. This part was deleted.

- Line 222. The use of “Conversely” should be rephrased. The numbers of newborn nymphs in both experiments are smaller than the control. The same holds true for “In contrast” on line 266.

Response: Thank you very much for the comment. We deleted “Conversely” in this point, and the latter sentence was deleted and replaced by a new one.

Appendix B

Associate Editor

Board Member

Comments to Author:

The referees appreciated the care and effort given to the revisions, and continue to recognize the importance of this study. At this stage, the reviewers still find some additional areas of the paper that need attention and revision, I recommend these be required before the paper can be accepted. These include revisions to Figure 3a, inclusion of sample sizes in Figures 1-2, several text clarifications, and some toning down of the interpretation so as to not over-generalize the implications of the results. I suggest the authors pay careful attention to these suggestions and address them fully. Another round of revisions will not be permitted, and some of these are important concerns and therefore it is imperative they are addressed thoroughly.

Response: Thank you very much for permitting us to revise the paper. We are very happy to read the comments by reviewers, and the comments are very helpful for improving the manuscript. We responded to all critical comments by reviewers, and revised the text following the suggestions and comments. We revised Abstract, Material and method, Results, and Discussion. We added one paper to References and statistical tests to Results.

We are very happy if this revision satisfies the requirements of the editors.

Responses to reviewers' comments

Reviewer 2

I cannot understand what the authors show in the new Fig. 3a. What does the strange minus number mean? Does it mean the authors also let the green clone start reproduction earlier than the yellow one? However, there is no remark in the materials and methods. Authors should indicate the number of the yellow clones produced during the first two or three days in the results, and also describe in the figure legends what the plotted samples represent.

Response: We are very sorry that our description made confusions. In Fig. 3a, we analyzed data by pooling the mixed colony experiments of simultaneous installation and time lag installation. In the horizontal axis of Fig. 3a, minus values mean the numbers of green aphid nymphs preexisting (we added this remark in Results, line 245). In the legend of Fig. 3a, we have already mentioned that “Minus numbers represent the numbers of green nymphs preexisting”. We also remarked this point in Results and in Material and methods, following the suggestion by the reviewer.

In Materials and methods, we added the following:

“We examined how the initial difference in the numbers of yellow and green nymphs affect the final sizes of these clones using all experiments of the simultaneous and time lag installation.”

In Results, we added the following:

“Yellow aphids produced on average 10.9 (± 3.33 SD) and 15.2 (± 4.15) nymphs for the first two and three days, respectively.”

Please indicate that the antennae-ablated aphids might not lose the ability to recognize self/nonself but to perceive crowding. They mention it very briefly in the text (Lines 253-254 and 331-332), but I do not think it is enough. This result cannot rule out the possibility that the antennae just perceive crowding, as the authors admitted in their Response. They should tone down their statements in the abstract and the discussion, which can be misleading.

Response: We agree the suggestion and so tried to tone down our statements.

In Abstract, we removed the phrase “using their antennae” from the sentence “Thus, we conclude that aphid clones can discriminate between self and non-self clones using their antennae,”.

Line 333-337. We added “or crowding” in the following sentence;

The fact that the three-day fecundity of an antennae-excised yellow aphid that was transferred to a highly dense green colony was not significantly different from that of a control yellow aphid suggests that some substances that can be detected by antennation are involved in the recognition of self and non-self clones “or crowding”.

Line 342-343. We newly added the following sentence;

However, the result can be explained if antennae-excised aphids lose the ability to perceive crowding [46].

Other comments.

- Please add sample sizes to Figures 1-2 or their legends.
- Lines 49 and 61. kin and non-kin clone -> closely related and unrelated clone?

Response: we have revised these points as follows,

- We added "For each curve, $n = 10$." to the legends of figures 1 and 2.
- In line 49, many authors use "kin and non-kin clone", so we kept it as it is.
- In line 61, we revised "kin and non-kin clone" to "closely related and unrelated clone".

Reviewer 1

- 1) You should better explain from line 125 why you performed an experiment where you introduce the green clone several days after the yellow one started to reproduce. Explain what is the rationale behind. Related to this experiment, I don't understand the figure 3. Why do you have negative numbers on the X axis? Why not simply give the number of yellow nymphs at the time you introduced the green clone. This is very confusing. Did you pool the results of two experiments (the one without time difference in the installation of the two clones and the one with)? Why did you draw a line to relate the datapoints? I doubt there is a linear relationship between pre-existing yellow nymphs and % of yellow aphids. It looks more there is a threshold function, with proportion of yellow aphids changing dramatically from 5 to 10 yellow nymphs. You should give more details in material and methods on how you analyse the data of this experiment. We discovered from lines 237 to 247 you performed a logistic regression and other stats (AIC) that are not presented. "Than when it was highly biased" is awkward. I guess you want to say "than for smaller or larger values". You also need to say briefly after line 247 what you conclude from these results.

Response: In lines 126-134 of Material and method, we explained in more detail why we performed an experiment where the green clone was introduced a few days after the yellow one started to reproduce by citing one paper. The paper (Mooney KA, Jones P,

Agrawal AA 2008) shows that the timing of colonization is critical to the outcome of competition between two aphid species. Thus, we think that it is important to consider the timing of reproduction as well as reproductive rates for competition between aphid clones.

We are very sorry that we made confusion in Fig. 3a. In this graph, we analyzed the results by pooling the experiments of the simultaneous installation and time lag installation. We added explanation in Materials and methods. In the experiment of the simultaneous installation, in some case, green aphids reproduced earlier (minus value), and in other cases, yellow aphids reproduced earlier (plus value).

We drew a regression curve based on logistic regression for the percentage of yellow aphids. This is because it is better to analyze percentage data using logistic regression. Linear regression is not suitable for percentage variables. We think that the dramatical change in the percentage is better described by logistic regression (actually, the curve may not well describe the sudden change).

Thank you very much for the suggestion. We changed “than when it was highly biased” to “than for smaller or larger values”.

We added a commentary after line 257 (present version) as follows;
“This result shows that aphids continue to reproduce when they are not inferior in number to their competitors.”

2) For the experiment using high density colonies, could you give the age range of the 130-150 aphids at the time you introduced the focal yellow aphid. Could you also give the age of the five aphids that are let to reproduce for 5 days. Did you remove all adults for both conditions (when 1 Y is introduced in high density Y and 1 Y in high density G) and at the same time? Please clarify.

Response: When we introduced the focal yellow aphid, the age range of the 130-150 aphids is from 1st to 3rd instar and at most 5-day old (except 5 adults), but it is difficult to calculate the proportions of the instars based on pictures. We transferred teneral adults to a new leaf, so on day 5, they are 14- or 15-day old from the birth. When we introduced 1Y in a high dense Y colony, we removed the 5 adults simultaneously. But, when we introduced 1Y in a high dense G colony, we did not remove the G adults.

3) Since you did not perform the reciprocal experiments (high density and antennae ablation) for the green clone, the result that aphids recognize self from non-self only stands for the yellow clone. This should be acknowledged in the discussion. Although I disagree with lines 270-272 since the green clone doesn't restrain reproduction in presence of the yellow one, at least not in the second reproductive period (from day 10). I really want the authors avoid generalizing their results as what is valid for the yellow clone is not for the green one, or has not been tested. It may be good in the discussion to compare the "strategies" of the two clones, which may rely on different cues and mechanisms.

Response: Following the suggestion, we mentioned that the discrimination of self from non-self clone only stands for the yellow clone in Discussion. We added some phrases in double quotes or in parenthesis as follows;

Line 315- The transfer experiment corroborated the hypothesis of reproductive restraint "only for the yellow clones".

Line 318- A high density of a given aphid clone ( the green clone) has a stronger negative effect on the reproduction of its competitor clone ( the yellow clone) than that of fellow clonal members.

In line 281-283, this statement (self-regulation of reproduction) is for clonal colonies only (starting from 2 clonal aphids) and is not related to mixed clone colonies, in which actually the green clone doesn't restrain reproduction in presence of the yellow one, as suggested by the reviewer. In the colonies of two clonal aphids, both the yellow clone and the green clone have the function of self-regulation, which is indicated by Fig 1 C and D.

We were not able to examine the reproductive response of one green aphid to a high density yellow colony, but we can fully predict the reproductive tactics of the green clone from the experiment in which yellow aphids reproduced a little earlier than did green aphids. When yellow aphids reproduced little earlier, green aphids restrained reproduction, suggesting that green aphids also have the ability of reproductive restraint.

4) Lines 212-213, test if significantly different and give p value.

Response: Thank you for the suggestion. we added the results of statistical tests as

follows;

(for mixed colonies vs. 2G, d.f. =1,18, $F = 80.7$, $p < 0.0001$; for mixed colonies vs. 2Y, d.f. =1,18, $F = 49.2$, $p < 0.0001$)

5) Line 290-293: I don't understand what you want to say and which experiments you are referring to.

Response: We refer to the result of figure 3b. So, we have already indicated (figure 3b) at the end of the sentence.

6) Line 304: which transfer experiment? Are you talking about the introduction of single aphids into high density colonies?

Response: Yes, it is. We revised it to "The transfer experiment of single aphids"

7) Line 314: "might be advantageous" and not "is advantageous" otherwise it's very speculative.

Response: We revised it to "might be advantageous".

8) Lines 318-320, since you removed the part on cuticular hydrocarbons, we have no evidence that the two clones differ in chemical substances. Please rephrase.

Response: Thank you for the suggestion. We excluded "chemical substances" and rephrased the sentences as follows;

There was evidence to suggest that the yellow clone was a dominant mutant (electronic supplementary material, Table S2) and had different phenotypes from the green clone with regard to body color and reproductive rates. The hypothesis that competing clones could detect the presence of each other through the antennation was tested in the present study by the antennae-excision experiment.

9) Lines 342-344, there are multiple examples of clonal mixing (different genotypes) on the same plant/colonies. Please rephrase.

Response: We cited seven papers (19-25) about clonal mixing in aphids.